# Novel App-Based Portable Spirometer for the Early Detection of COPD

**DOI:** 10.3390/diagnostics11050785

**Published:** 2021-04-27

**Authors:** Ching-Hsiung Lin, Shih-Lung Cheng, Hao-Chien Wang, Wu-Huei Hsu, Kang-Yun Lee, Diahn-Warng Perng, Hen-I. Lin, Ming-Shian Lin, Jong-Rung Tsai, Chin-Chou Wang, Sheng-Hao Lin, Cheng-Yi Wang, Chiung-Zuei Chen, Tsung-Ming Yang, Ching-Lung Liu, Tsai-Yu Wang, Meng-Chih Lin

**Affiliations:** 1Division of Chest Medicine, Changhua Christian Hospital, Changhua 500, Taiwan; teddy@cch.org.tw (C.-H.L.); shenghao@gmail.com (S.-H.L.); 2Institute of Genomics and Bioinformatics, National Chung Hsing University, Taichung 402, Taiwan; 3Department of Recreation and Holistic Wellness, MingDao University, Changhua 523, Taiwan; 4Department of Internal Medicine, Far Eastern Memorial Hospital, Taipei 220, Taiwan; 5Department of Chemical Engineering and Materials Science, Yuan Ze University, Zhongli, Taoyuan 320, Taiwan; 6Department of Internal Medicine, National Taiwan University Hospital, Taipei 100, Taiwan; haochienwang@gmail.com; 7Division of Pulmonary and Critical Care Medicine, Department of Internal Medicine, China Medical University Hospital, Taichung 404, Taiwan; wuhuei@gmail.com; 8Division of Pulmonary Medicine, Department of Internal Medicine, Shuang Ho Hospital, Taipei Medical University, New Taipei City 110, Taiwan; kangyenlee68@gmail.com; 9Department of Chest Medicine, Taipei Veterans General Hospital, Taipei 112, Taiwan; dwperng@vghtpe.gov.tw (D.-W.P.); chengyi@gmail.com (C.-Y.W.); 10Department of Internal Medicine, Cardinal Tien Hospital, Fu-Jen Catholic University, Taipei 242, Taiwan; heni@gmail.com; 11Department of Internal Medicine, Ditmanson Medical Foundation Chia-Yi Christian Hospital, Chiayi 600, Taiwan; mingshian@gmail.com; 12Division of Pulmonary and Critical Care Medicine, Department of Internal Medicine, Kaohsiung Medical University Hospital, Kaohsiung 807, Taiwan; jongrung@gmail.com; 13Division of Pulmonary and Critical Care Medicine, Department of Internal Medicine, Kaohsiung Chang Gung Memorial Hospital, Chang Gung University College of Medicine, Kaohsiung 833, Taiwan; chinchou@gmail.com; 14Division of Pulmonary Medicine, Department of Internal Medicine, National Cheng Kung University, College of Medicine and Hospital, Tainan 701, Taiwan; chiungzuei@gmail.com; 15Department of Pulmonary and Critical Care Medicine, Chang Gung Memorial Hospital, Chiayi Branch 613, Taiwan; tsungming@gmail.com; 16Division of Chest Medicine, Department of Internal Medicine, MacKay Memorial Hospital, Taipei 104, Taiwan; chinglung@gmail.com; 17Department of Thoracic Medicine, Chang Gung Memorial Hospital at Linkou, Chang Gung University, College of Medicine, Taipei 333, Taiwan; tsaiyu@gmail.com

**Keywords:** COPD, underdiagnosis, early detection, app-based spirometer

## Abstract

Chronic obstructive pulmonary disease (COPD) is preventable and treatable. However, many patients remain undiagnosed and untreated due to the underutilization or unavailability of spirometers. Accordingly, we used Spirobank Smart, an app-based spirometer, for facilitating the early detection of COPD in outpatient clinics. This prospective study recruited individuals who were at risk of COPD (i.e., with age of ≥40 years, ≥10 pack-years of smoking, and at least one respiratory symptoms) but had no previous COPD diagnosis. Eligible participants were examined with Spirobank Smart and then underwent confirmatory spirometry (performed using a diagnostic spirometer), regardless of their Spirobank Smart test results. COPD was defined and confirmed using the postbronchodilator forced expiratory volume in 1 s/forced vital capacity values of <0.70 as measured by confirmatory spirometry. A total of 767 participants were enrolled and examined using Spirobank Smart; 370 participants (94.3% men, mean age of 60.9 years and mean 42.6 pack-years of smoking) underwent confirmatory spirometry. Confirmatory spirometry identified COPD in 103 participants (27.8%). At the optimal cutoff point of 0.74 that was determined using Spirobank Smart for COPD diagnosis, the area under the receiver operating characteristic was 0.903 (95% confidence interval (CI) = 0.860–0.947). Multivariate logistic regression revealed that participants who have an FEV_1_/FVC ratio of <74% that was determined using Spirobank Smart (odds ratio (OR) = 58.58, 95% CI = 27.29–125.75) and old age (OR = 3.23, 95% CI = 1.04–10.07 for 60 ≤ age < 65; OR = 5.82, 95% CI = 2.22–15.27 for age ≥ 65) had a higher risk of COPD. The Spirobank Smart is a simple and adequate tool for early COPD detection in outpatient clinics. Early diagnosis and appropriate therapy based on GOLD guidelines can positively influence respiratory symptoms and quality of life.

## 1. Introduction

Chronic obstructive pulmonary disease (COPD) is a leading cause of morbidity and mortality worldwide [1,2]. In Taiwan, COPD is the seventh leading cause of death, with the corresponding age-adjusted mortality rates for men and women being 19.67 and 5.70 deaths per 100,000, respectively [3]. In general, COPD is treatable and preventable when identified in the early stage; nevertheless, the underdiagnosis of COPD remains a common challenge and imposes a considerable burden on healthcare systems and patients [4,5].

Spirometry is the gold standard for diagnosing COPD and monitoring treatment response [6]; however, it is still underused or unavailable in primary care settings or nonspecialized areas [4,7]. This may be attributed to various factors; for example, spirometry entails labor-intensive and time-consuming procedures and requires well-trained professionals for its execution [5,8]. Underuse of spirometry is considered a critical factor that is related to COPD underdiagnosis and results in unnecessary specialty referrals for diagnostic testing, creating barriers in COPD screening through spirometry and increasing the cost of diagnostic tests [4,9,10]. However, a report by the U.S. Preventive Services Task Force (USPSTF) revealed that the use of spirometry to screen for COPD in asymptomatic cases has no net benefit [11,12]. The early detection of COPD through a case-finding approach has been heavily advocated, and such an initiative provides opportunities for implementing interventions in the early stages, thus preventing disease progression [13,14]. Moreover, the USPSTF report suggested that COPD screening in at-risk populations has relatively high cost-effectiveness [15]. Hence, developing a feasible case-finding model that can be a suitable alternative to spirometry for identifying undiagnosed at-risk patients with COPD is urgently required.

A portable spirometer is a small and cheap device that can be a valid alternative to spirometry. Spirometers are advantageous for their ease of use, the requirement of less patient effort, and time-saving features, which render them useful for COPD screening [16,17,18,19]. Recently, the trend of integrating smart devices and portable medical devices has grown considerably and has affected the market and medical diagnosis field, including spirometry [20,21]. Systems integrating portable spirometers with smartphones can save processing power and reduce interface components, reduce medical device size and cost, and afford effective data communications [20,21,22]. However, only a few such systems have been validated for COPD screening, with their performance varying with the diagnostic testing procedure applied.

Spirobank Smart is an FDA-approved, app-based ultraportable device that connects to a smartphone app through Bluetooth for seamless recording of various lung function parameters, including the forced expiratory volume in one second (FEV_1_), PEF, forced vital capacity (FVC), FEV_1_/FVC, FEV_6_, and FEF_25–75_. Spirobank Smart can provide real-time feedback on the test quality, provide systematic and numeric visualizations of spirometer tests on the smartphone app, and afford effective data communication. A study conducted a correlation analysis between measurements obtained from Spirobank Smart and those obtained from confirmatory spirometry; the analysis indicated that this device exhibited acceptable validity and that it meets the latest ATS/ERS standards for accuracy [23]. However, the performance of Spirobank Smart in COPD screening remains unknown.

Accordingly, to fill the aforementioned knowledge gap, we conducted this study to investigate the feasibility of Spirobank Smart for the early detection of COPD. Specifically, we applied Spirobank Smart to assist in the early detection of COPD using case-finding in outpatient clinics. We determined the correlation between FEV_1_/FVC values obtained by Spirobank Smart and postbronchodilator (post-BD) FEV_1_/FVC ratio obtained through confirmatory spirometry (executed using a diagnostic spirometer). Furthermore, we determined the cutoff point for the FEV_1_/FVC ratio and the corresponding predictive performance of Spirobank Smart in identifying COPD in high-risk populations; subsequently, we evaluated whether the FEV_1_ obtained using Spirobank Smart was useful for classifying patient disease severity according to the GOLD (Global Initiative for Chronic Obstructive Lung Disease) severity stratification system.

## 2. Materials and Methods

### 2.1. Study Design

This prospective multicenter validation study was conducted for the period from April 2018 to December 2019, and the study protocol was approved by the Institutional Review Board of Changhua Christian Hospital (IRB number 190806); Institutional Review Board of Changhua Christian Hospital reviewed the project and deemed it to be exempt research because the evaluation was for a public service program and all data were de-identified. The requirement for written consent was waived. Participants were enrolled from 26 outpatient clinics in Taiwan, from which their demographic information, CAT (COPD Assessment Test) questionnaires, Spirobank Smart measurements, and diagnostic spirometer measurements were obtained.

### 2.2. Study Participants

The Taiwan Society of Pulmonary and Critical Care Medicine recruited participants and implemented a study, commissioned by the Taiwan Ministry of Health and Welfare, on early diagnosis of COPD. The eligibility criteria were as follows: having an age of ≥40 years, having ≥10 pack-years of smoking, exhibiting chronic respiratory symptoms (cough, phlegm, or dyspnea, or combination thereof), and not having a confirmed COPD diagnosis within the prior year. We excluded individuals who did not undergo post-BD spirometry because we could not confirm whether a given individual had a COPD diagnosis; we also excluded those who were unable to correctly operate the Spirobank Smart device. Accordingly, a total of 370 participants completed all the tests, and their data were analyzed in this study. A flowchart of the participant enrollment procedure is presented in Figure 1.

### 2.3. Study Procedures

All tests undertaken in the 26 outpatient settings were conducted on the same day. The participates were assessed using Spirobank Smart and required to complete a CAT questionnaire. Regarding the operation of the Spirobank Smart device, each nurse and physician who contributed to this study were adequately trained on the use of the device. The participants were also required to inhale maximally and then exhale forcibly into the mouthpiece of the device for at least 6 s until the chronometer was switched on and changed its color to green. Measurements of acceptable quality, that is those with the highest summed value (FEV_1_ + FVC), were subsequently recorded; individual ratios were then calculated. Cross-contamination was minimized using disposable plastic mouthpieces. Spirometry with BD reversibility was performed independently by trained operators in accordance with the guidelines proposed by the American Thoracic Society. Lung function parameters were measured before and after 20 min of BD inhalation. At least three adequate baseline and post-BD FVC maneuvers were performed to measure the highest FEV_1_ and FVC values. COPD diagnosis and COPD severity classification were conducted in accordance with GOLD definitions. The reference criterion for COPD was defined as a post-BD FEV_1_/FVC value of <70%, and COPD severity was classified according to the post-BD FEV_1_ percentages of the predicted values.

### 2.4. Devices and System

The Spirobank Smart device (MIR, Rome, Italy) used in this study can connect to smartphone apps through Bluetooth for the seamless recording of lung function parameters; it makes measurements through a bidirectional digital turbine. The turbine sensor operates based on the infrared interruption principle, which can ensure appropriate and repeatable measurement procedures. The device does not require calibration and applies a disposable turbine flow meter for its measurements. In this study, each participant’s demographic information, including age, gender, height, and weight, was manually inputted into the device before the spirometry could be performed. To obtain measurements using this spirometer, each participant was required to exhale into the turbine, thus activating the motor inside the spirometer; concurrently, the speed of the rotor was recorded, and the recorded data were then adapted and transmitted to the app on the participant’s smartphone. The exhalation process switched on the chronometer, and the color of the chronometer changed from orange to green after 6 s of exhalation. Accordingly, data on several parameters, including FEV_1_, PEF, FVC, FEV_1_/FVC, FEV_6_, and FEF_25–75_, were instantly displayed on the app. When an error occurred, the device detected and indicated the error description. Each participant was required to perform three cycles of inhalation and exhalation, and the best FEV_1_ and FVC values were selected. GLI-2012-predicted values were used as the reference values (as reported in our previous study [24]) for the executed spirometry. Such predicted values can be expressed as follows: predicted value = e^a^ × H^b^ × A^c^ × e^d×group^ × e^spline^. In the preceding equation, “group” represents Southeast Asian.

### 2.5. Statistical Analysis

Data are expressed as a frequency with a percentage and as mean ± standard deviation for categorical and continuous variables, respectively. The distributions of the variables between COPD and non-COPD were compared using Student’s *t*-test and the chi-square test. The agreement between the post-BD FEV_1_/FVC values measured through confirmatory spirometry and pre-BD FEV_1_/FVC values measured using Spirobank Smart was assessed using Bland–Altman plots. The optimal cutoff point for the FEV_1_/FVC ratio determined by Spirobank Smart for identifying COPD in high-risk participants was determined using the Youden index derived from receiver operating characteristic (ROC) analysis. Sensitivity, specificity, positive predictive values (PPVs), negative predictive values (NPVs), and ROC curve values were then used to assess the effectiveness of the aforementioned screening tools in differentiating between COPD and non-COPD. Logistic regression analyses were performed to determine the association between potential risk factors and COPD incidence. All statistical analyses were performed using IBM SPSS 22 (IBM Corp., Armonk, NY, USA). In all analyses, two-tailed *p*-values < 0.05 were considered statistically significant.

## 3. Results

### 3.1. Demographic Characteristics of Study Participants

This study included data from 26 hospitals and clinics in Taiwan, including medical centers, regional hospitals, district hospitals, and clinics. The demographic characteristics of the enrolled participants are listed in Table 1. Among the participants, 103 had COPD (COPD group) and 267 did not have COPD (non-COPD group); of the participants, 94.3% were men. Participants with COPD appeared to be older, have higher pack-years of smoking, and have lower body mass index (BMI) values. The pre-BD FEV_1_/FVC ratio that was determined using Spirobank Smart was significantly lower in the COPD group (63.57 ± 13.37) than it was in the non-COPD group (81.78 ± 7.44; *p* < 0.001). The COPD group had a significantly higher CAT score (12 ± 7) and lower post-BD FEV_1_/FVC ratio (59.06 ± 9.04) than did the non-COPD group (CAT score: 9 ± 6; post-BD FEV_1_/FVC ratio: 81.28 ± 7.19; all *p* < 0.001). These variables demonstrated statistically significant differences between the two groups.

### 3.2. Agreement between Post-BD FEV_1_/FVC Ratios Measured Using the Confirmatory Spirometry and Pre-BD FEV_1_/FVC Ratios Measured Using Spirobank Smart

Bland–Altman plots (Figure 2) are useful for determining the relationship between differences and averages, which can help researchers to explore any systematic biases and identify possible outliers. In this study, we derived Bland–Altman plots to evaluate the agreement between the pre-BD FEV_1_/FVC ratios measured using Spirobank Smart and the post-BD FEV_1_/FVC ratios measured using the confirmatory spirometry. The plots revealed that few values (5.67%) fell outside the 95% confidence interval. The mean difference between the post-BD values derived using the confirmatory spirometry and the FEV_1_/FVC values derived using Spirobank Smart was 1.6%. Additionally, the plots demonstrated that the limits of agreement were superior (narrower) for the mean of FEV_1_/FVC values of <70% than they were for the mean of FEV_1_/FVC values of ≥70%.

### 3.3. ROC Curves and Diagnostic Accuracy for the Pre-BD FEV_1_/FVC Ratios Measured Using Spirobank Smart

We constructed ROC curves (Figure 3) to assess the suitability of FEV_1_/FVC measured using Spirobank Smart as a prescreening measure for COPD identification. Our results indicated that FEV_1_/FVC measured by Spirobank Smart could significantly predict COPD, with the corresponding area under the ROC curve (AUROC) being 0.903 (95% CI = 0.860–0.947). According to the Youden index derived from our ROC analysis, Spirobank-Smart-measured FEV_1_/FVC values of < 74% can serve as high-risk indicators of COPD, and the highest predictive ability was observed at an AUROC value of 0.873 (95% CI = 0.827–0.920).

The cutoff values for FEV_1_/FVC and the corresponding predictive performance of Spirobank Smart are presented in Table 2. As indicated in this table, at the optimal cutoff point of 0.74, the sensitivity, specificity, PPV, and NPV were 82.50%, 92.13%, 80.20%, and 93.20%, respectively. Compared with the sensitivity of Spirobank Smart with an FEV_1_/FVC ratio of <70, the sensitivity of the diagnostic spirometer was as low as 70.90%; thus, most patients with COPD were underdiagnosed, despite the AUROC being 0.834 (95% CI = 0.779–0.889). The portable spirometer achieved a balance between sensitivity and specificity when the FEV_1_/FVC ratio was <74%; accordingly, it exhibited the best predictive ability. Our findings revealed that Spirobank Smart could be used as a diagnostic tool in general population screening procedures for identifying patients that are at high risk of COPD.

Figure 4 illustrates the diagnostic performance, as assessed with respect to COPD severity, that was observed at the optimal cutoff point for FEV_1_/FVC (<74%). Some of the participants without COPD were recorded as having COPD, resulting in false-positive results; this thus led to moderate PPV estimates (80.20%). In addition, relatively few participants with COPD were recorded as not having COPD, resulting in false-negative results; this thus led to high NPV estimates (93.20%). Of the participants for whom false-negative results were recorded, 80% had mild or only moderate COPD.

### 3.4. GOLD Classification and CAT Score of Participants Based on the FEV_1_ Values Obtained Using Confirmatory Spirometry and Spirobank Smart

To evaluate whether the FEV_1_ determined using Spirobank Smart could be a reliable parameter for classifying obstruction severity, we analyzed the obstruction severity by using the FEV_1_ values obtained through confirmatory spirometry and those determined using Spirobank Smart; the analysis results are summarized in Table 3. The FEV_1_ values obtained using the diagnostic spirometer revealed that 24.3% and 51.5% of the participants exhibited mild and moderate COPD (GOLD I and II), respectively. The classification results obtained using the FEV_1_ values determined using Spirobank Smart were comparable to those obtained using the values determined using the diagnostic spirometer (i.e., 15.5% and 57.3% of the participants were classified as having GOLD I and II severity levels, respectively). Moreover, the CAT scores for grades III and V were higher than those for grades I and II.

### 3.5. Associations of FEV_1_/FVC Determined Using Spirobank Smart and the Participant Characteristic Variables with the COPD Incidence

Univariate logistic regression analysis revealed that factors such as an FEV_1_/FVC ratio of <74% that was determined using Spirobank Smart, ≥50 pack-years of smoking, age, and lower BMI were positively associated with the incidence of COPD. After the multivariate adjustments, we observed that an FEV_1_/FVC ratio of <74% that was determined using Spirobank Smart (odds ratio (OR) = 58.58, 95% CI = 27.29–125.75) and old age (OR = 3.23, 95% CI = 1.04–10.07 for 60 ≤ age < 65; OR = 5.82, 95% CI = 2.22–15.27 for age ≥ 65) remained significantly associated with the incidence of COPD (Table 4).

## 4. Discussion

Earlier detection of COPD in patients can improve the short- and long-term patient outcomes when treated using current therapies. Although spirometry is the gold standard for COPD diagnosis, it is often perceived as an expensive and time-consuming process, leading to the underdiagnosis or misdiagnosis of COPD. Furthermore, spirometry is not cost-effective for screening for COPD in a population of truly asymptomatic smokers. To the best of our knowledge, few efficacious strategies have been designed for identifying patients with undiagnosed COPD who are most likely to benefit from current therapies, especially in Taiwan [19]. Accordingly, to bridge this gap, we used an app-based portable spirometer, namely, Spirobank Smart, to identify undiagnosed COPD in at-risk populations as the first step in determining which patients should be referred for further COPD diagnostic evaluation.

We validated the feasibility and suitability of Spirobank Smart for deriving FEV_1_/FVC values for screening undiagnosed patients who were at risk of COPD. Based on our knowledge, this is the first study in Taiwan to evaluate the accuracy and feasibility of an app-based spirometer for early COPD detection. The principal findings of this study regarding at-risk population outcomes are outlined as follows. First, we found that 27.8% of the participants were newly diagnosed with COPD using confirmatory spirometry at tertiary hospitals, and nearly 70% of the newly diagnosed patients had mild or moderate COPD. Second, the AUROC value for COPD identification using Spirobank-Smart-derived was 0.903. Third, patients who have an FEV_1_/FVC ratio of <74% that was determined using Spirobank Smart (odds ratio (OR) = 60.70, 95% CI = 27.95–131.83) and old age (OR = 3.23, 95% CI = 1.04–10.07 for 60 ≤ age < 65; OR = 5.82, 95% CI = 2.22–15.27 for age ≥ 65) exhibited a higher risk of COPD; this indicates that COPD incidence was associated with increased age and FEV_1_/FVC ratio < 74% determined by Spirobank-Smart. These findings support the claim that Spirobank Smart is an acceptable and feasible screening tool for the early detection of COPD.

A previous study reported a COPD prevalence of 18.9% when respiratory symptoms were not considered in the inclusion criteria [25]. When respiratory symptoms were considered in the inclusion criteria, the prevalence of COPD increased to 25–52.9% [16,18]. In Taiwan, a nationwide survey of the general population revealed an estimated COPD prevalence of 6% [26]; however, a case-finding study reported that the COPD prevalence in an at-risk population was 48.8% [27]. The present study indicated a COPD prevalence of 27.85% in smokers who were aged >40 years and had ≥10 pack-years of smoking and at least one symptom. Kjeldgaard et al. used a similar study design and similar inclusion criteria to those of the present study; they determined the COPD prevalence to be 32% in an at-risk population (aged ≥40 years with smoking history and at least one respiratory symptom) [18]. Kim et al. measured the COPD prevalence in smokers who had 10 pack-years of smoking and were aged >40 years in a primary care setting; they reported that the prevalence of COPD was 23.7% [28]. These findings suggest that the prevalence would be significantly influenced by the inclusion criteria that are used to define COPD and that screening for high-risk individuals can detect high proportions of patients that are undiagnosed as having COPD. Furthermore, our findings demonstrate that Spirobank Smart could be used for early detection of COPD in an outpatient clinical setting, as evidenced by a follow-up diagnostic spirometer test used for confirmation; the concomitant presence of respiratory symptoms increased the likelihood of identifying COPD.

In the present study, the level of agreement between the pre-BD FEV_1_/FVC values that were determined using Spirobank Smart and the post-BD FEV_1_/FVC values that were determined using the diagnostic spirometer was satisfactory; this thus demonstrates that Spirobank Smart is an acceptable alternative screening tool for early detection of COPD. Notably, the limits of agreement for the FEV_1_/FVC values of <70% were narrower than those for FEV_1_/FVC values of ≥70%, implying that FEV_1_/FVC values of ≥70% may lead to slight underestimations. In other words, the underdiagnosis of obstruction may occur in younger adults or those with mild COPD. Therefore, we suggest that FEV_1_/FVC values of <0.74, instead of FEV_1_/FVC values of  <0.70 (the conventional GOLD definition), be used along with Spirobank Smart for optimal screening; additionally, this cutoff resulted in the achievement of a balance between false-positive and false-negative results in our study population, thereby reducing the possibility of underdiagnosis or misdiagnosis.

Few reports are available on the feasibility and reliability of portable devices for COPD screening in clinical settings. Frith et al. evaluated the feasibility of piko-6 in the early detection of COPD without including a BD test [29]. Two other studies have also used COPD-6 and piko-6 screening devices. These three studies have indicated that COPD-6 and piko-6 implemented with FEV_1_/FEV_6_ for COPD screening in at-risk populations had AUROC values of 0.75 and 0.86, respectively, validating these screening devices as acceptable tools for COPD screening in at-risk populations [16,17,18,19,29]. These findings are similar to those of the present study. Although FEV_6_ is an acceptable surrogate for FVC in spirometry, FEV_6_ has some drawbacks, such as the underestimation of mild airway obstruction [30,31,32]. Hernández et al. reported that FEV_1_/FVC values of <0.7 determined using a smartphone-based spirometer, namely Air-Smart, for detecting obstructive airway diseases had a sensitivity of 94.0% and a specificity of 97.2% [20]. Although the use of portable devices to diagnose COPD or obstructive airway diseases has been validated, the results cannot be compared with those of the current study due to the differences in experimental design and spirometry procedures between the studies.

ROC analysis for the FEV_1_/FVC values that were determined using Spirobank Smart demonstrated that this parameter could differentiate between the COPD and non-COPD groups with moderate to good AUROC values (0.903 (95% CI = 0.860–0.947)). The optimal cutoff point (FEV_1_/FVC value of <74.0%) with a score of 0.757 was determined according to the Youden index. A screening method with perfect sensitivity, specificity, PPV, and NPV is not usually available. In COPD screening, high sensitivity, rather than high specificity, is prioritized because identifying more potential patients with COPD-like conditions is crucial. The PPVs derived in this study revealed that approximately 20% of patients had post-BD FEV_1_/FEV values below the optimal cutoff point of <0.7; therefore, the PPV is useful for avoiding unnecessary, excessive examination of participants due to false positives. Additionally, high NPVs were observed in this study, indicating that 90% of patients had FEV_1_/FVC values of ≥74.0%, as determined using Spirobank Smart; therefore, we could feasibly determine whether a participant could have COPD and require further testing. Our results are similar to those of a previous study that examined the COPD screening accuracy of a handheld spirometer according to the AUROC, sensitivity, specificity, PPV, and NPV at a specified cutoff point. Additional studies are warranted to determine the clinically sensible cutoff point of the parameter in Spirobank Smart in other cohorts and thereby confirm its clinical usefulness for COPD screening. The use of a spirometer alongside a BD is the gold standard for COPD diagnosis. However, a BD is not sufficiently safe for implementation in primary clinics or nonspecialist areas due to insufficient monitoring and emergency procedures [28]. Therefore, we suggest that for individuals who are at risk of COPD and have abnormal testing results from Spirobank Smart, another diagnostic procedure executed using a spirometer alongside a BD should be used. Accordingly, Spirobank Smart may reduce inappropriate BD use in COPD diagnoses and provide a feasible screening process.

The conventional GOLD definition may lead to the underdiagnosis of obstruction in younger adults and overdiagnosis of obstruction in older adults [33,34]. Therefore, scholars have suggested the use of a lower limit of normal (LLN)-based diagnosis of COPD [33,34,35]. However, a previous study reported that LLN-based definitions tend to underdiagnose COPD in symptomatic patients [4]. In our study design, we used a case-finding strategy based on symptom screening. The conventional GOLD definition may, therefore, be the appropriate choice for reducing the underdiagnosis of COPD in our screening strategy. In addition, previous studies have indicated that the LLN-based diagnosis of COPD generated fewer false positives and more false negatives compared with the conventional GOLD definition. False negatives lead to the undertreatment of patients with COPD during disease stages (e.g., GOLD I and II) when they are likely to benefit most. Moreover, LLN tends to categorize elderly adults with mild obstruction into a non-COPD category (due to its high specificity and low sensitivity) [36]. In our study population, the mean age was 60.9 years, and a large proportion of the participants had COPD in GOLD I and II; thus, the use of the conventional GOLD definition may help to avoid potential false-negative results. A previous study revealed that compared with diagnoses executed using fixed cutoff values, diagnoses based on the LLN involved slightly higher NPVs [37]. Considering the potential false positives that can result from using the conventional GOLD definition and considering the evolving understanding of strategies for airflow obstruction detection, the use of the LLN-based definitions as a gold standard in spirometry to validate the FEV_1_/FVC values that were determined using Spirobank Smart could be valuable and warrants further study.

A CAT score of ≥10 points is used to classify patients with COPD as highly symptomatic. In the present study, 41.7% of patients that were newly diagnosed as having COPD had CAT scores of <10, which indicated that the presence of fewer symptoms may contribute to diagnostic confusion and clinical neglect. Moreover, studies have reported that handheld spirometers are useful for determining obstruction severity. In our study, we assessed the feasibility of evaluating COPD severity in an at-risk population by using FEV_1_. We determined that nearly 70% of the newly diagnosed COPD cases were classified as GOLD I or II based on the FEV_1_ values that were measured using Spirobank Smart. The frequency of GOLD classifications was similar to that of the diagnostic spirometer. Our findings also demonstrate that the prevalence of COPD in GOLD I was 24.3%, which is similar to that (27.2%) observed in a previous study conducted at a single medical center in Taiwan [23]; this suggests that Spirobank Smart is useful for identifying undiagnosed COPD with mild or moderate airway limitations in at-risk populations and can help to raise awareness of current perceptions of COPD. Based on our study findings, to provide the appropriate intervention, we propose the following clinical procedure for the triage of participants who are at risk of COPD: (1) if the FEV_1_/FVC ratio is <0.74, the patient should be referred to a pulmonologist for further diagnosis; (2) if the FEV_1_/FVC ratio is ≥0.74, a diagnosis alternative to COPD should be considered, and smoking cessation should be recommended to the patient; (3) if a patient is newly diagnosed as having COPD, they should be invited to join the COPD pay-for-performance program that is implemented by the National Health Insurance Administration to encourage physicians to provide patient-centered care plans; (4) if an individual is diagnosed as not having COPD, they should be advised to cease smoking (Figure 5). Furthermore, our findings indicated that patients with known smoking history who are old age had a higher risk of COPD. The English Longitudinal Study of Ageing (ELSA) demonstrated that nearly all smoking-related deaths (99%) were due to the occurrence of COPD in people aged >50 years, and the mean duration of smoking for current smokers was 42 years [38]. These findings suggest that an age of >40 years and a smoking history of ≥10 pack-years are adequate criteria for early COPD detection. In our study, we believe that symptomatic individuals who are aged >60 years with smoking history have greater risk of COPD and deserve closely routine COPD screening.

Limitations regarding the implementation of Spirobank Smart should be acknowledged. The data were collected from only a limited number of participants enrolled in hospital-based facilities, including medical centers and regional hospitals, which may not reflect the entire COPD population, especially those who have never visited hospitals for COPD testing. Additional studies including more types of medical facilities, such as primary care facilities, are warranted to assess the validity of Spirobank Smart. Moreover, the FEV_1_ measurements that were obtained using Spirobank Smart were underestimated by up to 5%, and the FEV_1_/FVC ratios were underestimated by 3–4%. These underestimations may cause by the inaccuracy of the portable device. However, Spirobank Smart complies with the latest documented ATS/ERS standards for accuracy, and our ROC analysis indicated that Spirobank Smart was sufficiently accurate for COPD identification through the case-finding strategy. Finally, the interday repeatability of the lung function test, even in the clinical trial population, was determined to be considerable, which may be a limitation of our study. A previous study highlighted the obscurity of lung function values from a single spirometry procedure. The results indicated that approximately 1% of the participants’ values changed from the lowest to highest quintiles (and vice versa), and only approximately half of the participants allocated to different lung function quintiles at screening were grouped into the same quintile at baseline. A possible explanation for this is the methodological variability of lung function measurements and the physiological variability of the airway caliber [39].

In conclusion, Spirobank Smart was determined to be a simple and feasible device for screening undiagnosed COPD in at-risk populations in outpatient clinical settings. We observed that measurements obtained from the device had a moderate correlation with those obtained from a diagnostic spirometer and that the device had acceptable accuracy in identifying undiagnosed COPD in our at-risk population. The use of an app-based spirometer is a potential strategy for improving the early detection of COPD. We suggest that individuals who are aged >40 years and have >10 pack-years of smoking and at least one respiratory symptom should receive a lung function evaluation using an app-based spirometer. Furthermore, we suggest that participants who have abnormal test results, as determined using Spirobank Smart with FEV_1_/FVC values of <0.74, should undergo diagnostic spirometrer to confirm the COPD diagnosis and initiate an appropriate intervention in the early stages of the disease.

## Figures and Tables

**Figure 1 diagnostics-11-00785-f001:**
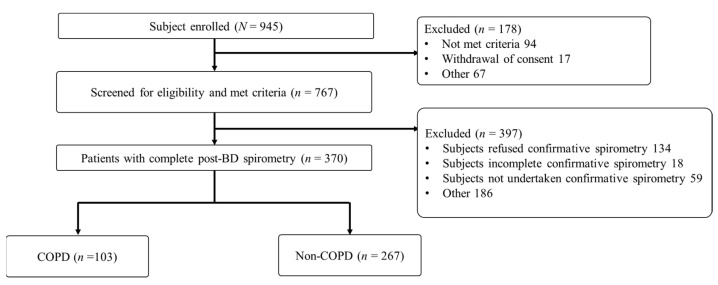
Flowchart of participant selection for the early detection of chronic obstructive pulmonary disease (COPD).

**Figure 2 diagnostics-11-00785-f002:**
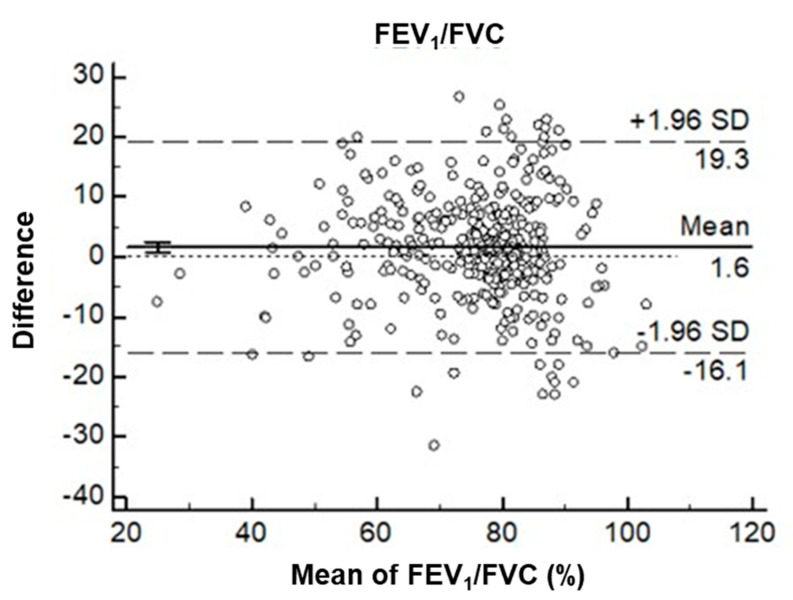
Bland–Altman plots illustrating the differences between the prebronchodilator (pre-BD) forced expiratory volume in 1 s (FEV_1_)/forced vital capacity (FVC) values obtained using Spirobank Smart and the post-BD FEV_1_/FVC values obtained using the diagnostic spirometer as a percentage of the mean difference (vertical axis) versus the mean of the two FEV_1_/FVC ratios (horizontal axis).

**Figure 3 diagnostics-11-00785-f003:**
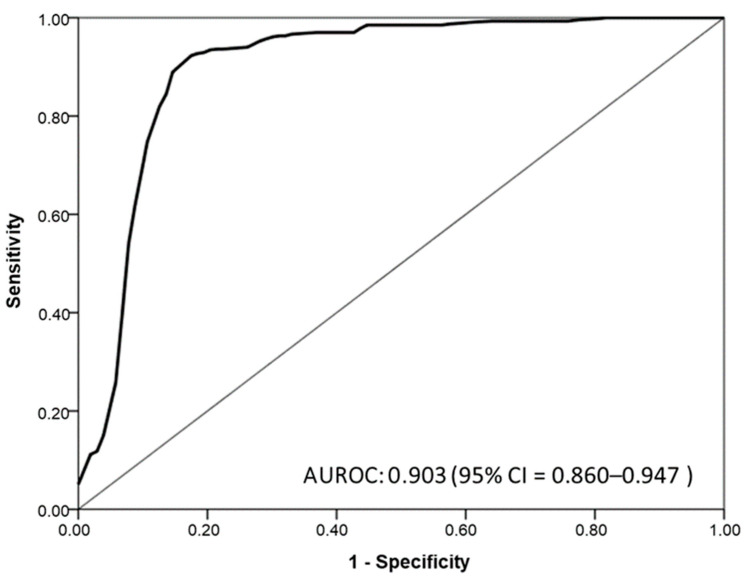
Receiver operating characteristic (ROC) curves for the forced expiratory volume in 1 s (FEV_1_)/forced vital capacity (FVC) ratio measured using Spirobank Smart.

**Figure 4 diagnostics-11-00785-f004:**
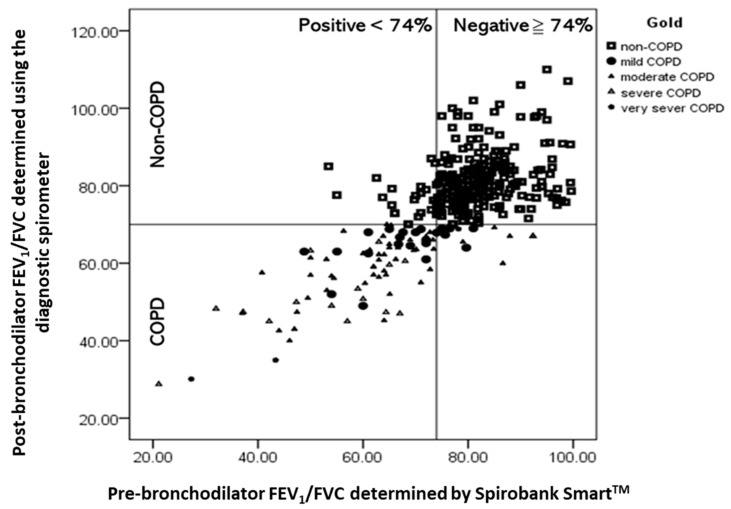
Scatter plots of post-bronchodilator (post-BD) forced expiratory volume in 1 s (FEV_1_)/forced vital capacity (FVC) ratios that were determined using the diagnostic spirometer against pre-BD FEV_1_/FVC ratios that were determined using Spirobank Smart for eligible participants. The quadrants are defined by the GOLD spirometer criteria for obstruction (post-BD FEV_1_/FVC < 70%) and the Spirobank-Smart-measured FEV_1_/FVC ratios that yielded optimal performance characteristics (FEV_1_//FVC < 74%).

**Figure 5 diagnostics-11-00785-f005:**
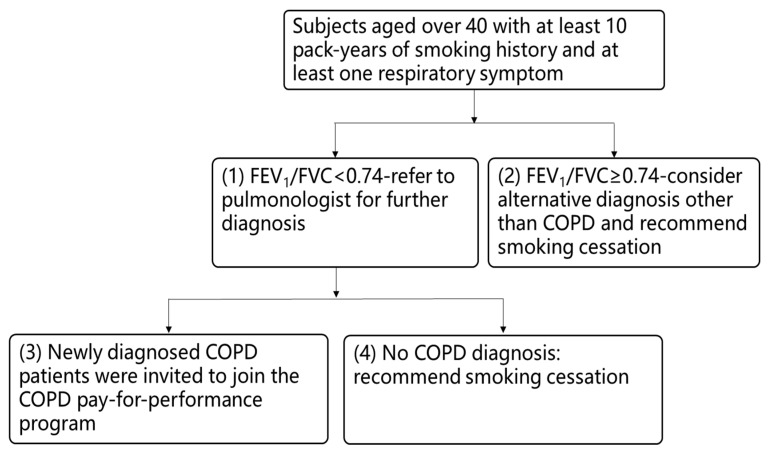
Clinical procedure for the triage of participants who are at risk of chronic obstructive pulmonary disease (COPD) based on the results obtained in this study using a lung function test and Spirobank Smart.

**Table 1 diagnostics-11-00785-t001:** Demographic characteristics of eligible participants for the early detection of chronic obstructive pulmonary disease (COPD).

Characteristics	Non-COPD	COPD	Total	*p*-Value
Sample size	267	103	370	-
Age (mean ± SD)	59.0 ± 9.0	65.7 ± 9.8	60.9 ± 9.7	<0.001
<55 years	93 (34.8%)	17 (16.5%)	110 (29.7%)	<0.001
55–59 years	46 (17.2%)	9 (8.7%)	55 (14.9%)	
60–64 years	56 (21%)	17 (16.5%)	73 (19.7%)	
≥65 years	72 (27%)	60 (58.3%)	132 (35.7%)	
Gender				
Male	253 (94.8%)	96 (93.2%)	349 (94.3%)	0.536
Female	14 (5.2%)	7 (6.8%)	21 (5.7%)	
BMI	25.81 ± 3.86	24.38 ± 4.13	25.41 ± 3.98	0.001
Cough				
No	24 (9.0%)	4 (3.9%)	28 (7.6%)	0.096
Yes	243 (91.0%)	99 (96.1%)	342 (92.4%)	
Phlegm				
No	30 (11.2%)	9 (8.7%)	39 (10.5%)	0.483
Yes	237 (88.8%)	94 (91.3%)	331 (89.5%)	
Breathless				
No	91 (34.1%)	24 (23.3%)	115 (31.1%)	0.045
Yes	176 (65.9%)	79 (76.7%)	255 (68.9%)	
CAT	9 ± 6	12 ± 7	10 ± 6	
0–9	160 (59.9%)	43 (41.7%)	203 (54.9%)	0.001
10–19	94 (35.2%)	44 (42.7%)	138 (37.3%)	
20–29	12 (4.5%)	15 (14.6%)	27 (7.3%)	
30–40	1 (0.4%)	1 (1.0%)	2 (0.5%)	
Smoking pack-years	39.4 ± 27.5	48.6 ± 29.3	42.6 ± 28.3	0.001
<50	216 (80.9%)	67 (65.0%)	283 (76.5%)	<0.001
≥50	51(19.1%)	36 (35.0%)	87 (23.5%)	
Pre-bronchodilator FEV_1_/FVC determined using Spirobank Smart	81.78 ± 7.44	63.57 ± 13.37	76.71 ± 12.49	<0.001
Post-bronchodilator FEV_1_/FVC determined using a diagnostic spirometer	81.28 ± 7.19	59.06 ± 9.04	75.1 ± 12.62	<0.001

**Table 2 diagnostics-11-00785-t002:** Diagnostic accuracy of Spirobank Smart across different cutoff points.

Device Cutoff Ratio (FEV_1_/FVC)	Sensitivity (%)	Specificity (%)	PPV (%)	NPV (%)	AUROC (95% CI)
<70%	70.90%	95.88%	86.90%	89.50%	0.834 (0.779–0.889)
<71%	73.80%	94.01%	82.60%	90.30%	0.839 (0.786–0.892)
<72%	77.70%	93.63%	82.50%	91.60%	0.857 (0.806–0.907)
<73%	80.60%	92.88%	81.40%	92.50%	0.867 (0.819–0.916)
<74%	82.50%	92.13%	80.20%	93.20%	0.873 (0.827–0.920)
<75%	86.40%	84.64%	68.50%	94.20%	0.855 (0.810–0.901)
<76%	87.40%	82.02%	65.20%	94.40%	0.847 (0.801–0.893)

**Table 3 diagnostics-11-00785-t003:** GOLD classifications and CAT scores of the patients. The scores depended on the forced expiratory volume in 1 s (FEV_1_) values obtained using a diagnostic spirometer and Spirobank Smart. The characteristics of patients with chronic obstructive pulmonary disease (COPD) were defined according to the GOLD classification of disease severity.

GOLD Grade	Diagnostic Spirometer		Spirobank Smart	
	*n* (%)	CAT Score	*n* (%)	CAT Score
GOLD I	25 (24.3%)	10 ± 5	16 (15.5%)	10 ± 4
GOLD II	53 (51.5%)	12 ± 7	59 (57.3%)	11 ± 6
GOLD III	22 (21.3%)	11 ± 6	22 (21.4%)	13 ± 8
GOLD IV	3 (2.9%)	21 ± 19	6 (5.8%)	18 ± 12
Total	103 (100%)	12 ± 7	103 (100%)	12 ± 7

**Table 4 diagnostics-11-00785-t004:** Multivariate logistic regression analysis used to evaluate the associations between the forced expiratory volume in 1 s (FEV_1_)/forced vital capacity (FVC) that was determined using Spirobank Smart, patients characteristic variables, and chronic obstructive pulmonary disease (COPD) incidence.

Variables	Crude OR (95% CI)	*p*-Value	Adjusted OR (95% CI)	*p*-Value
Portable spirometer: FEV1/FVC< 74%	55.14 (28.13–108.77)	<0.001	58.58 (27.29–125.75)	<0.001
Smoking PY ≥ 50	2.28 (1.37–3.78)	0.001	1.31 (0.57–2.98)	0.535
Age category				
Age < 55	1		1	
55 ≤ age < 60	1.07 (0.44–2.59)	0.864	1.12 (0.32–3.886)	0.864
60 ≤ age < 65	1.66 (0.79–3.51)	0.185	3.23 (1.04–10.07)	0.04
Age ≥ 65	4.56 (2.45–8.48)	<0.001	5.82 (2.22–15.27)	<0.001
CAT category				
0–9	1		1	
10–19	1.74 (1.07–2.85)	0.027	1.39 (0.65–2.98)	0.393
20–29	4.65 (2.03–10.67)	<0.001	3.43 (0.99–11.29)	0.052
30–40	3.72 (0.23–60.71)	0.356	5.89 (0.06–613.56)	0.535
BMI	0.907 (0.85–0.97)	0.002		
Gender (male)	0.759 (0.30–1.94)	0.759		

## Data Availability

The data presented in this study are available on request from the corresponding author. The data are not publicly available due to ethical reasons.

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
