# Peer review of "Novel App-Based Portable Spirometer for the Early Detection of COPD"

_diagnostics, 2021, doi:10.3390/diagnostics11050785_

Round 1

Reviewer 1 Report

Thank you for giving me the opportunity to review the research by Ching-Hsiung et al. This is a study evaluating the feasibility of an app-based spirometer for the early detection of COPD among the risk group of symptomatic smokers. As COPD is common in the general population, affecting around 5 to 10% of the human population, this is an interesting topic for the readers and public health stakeholders.

The study is well structured and designed. The authors’ work provides conclusions that a portable app-based spirometer is a potential strategy to improve early detection of COPD. Study limitations are recognized. I have no major comments related to the authors' work and manuscript reporting its results. However, the manuscript requires minor revisions before possible publication:

  1. The English language of the manuscript requires significant improvement. The text would benefit from revision by the native speaker and English language medical writing corrections.
  2. Authors should report what spirometric reference values were used by app-based and confirmatory spirometer?
  3. The authors could try to analyze their data according to the lower limit of normal (LLN) for FEV1/FVC to diagnose COPD – despite recommended by GOLD guidelines, a fixed cut-off of 70% may lead to underdiagnosis of obstruction in the younger population and overdiagnosis of obstruction in the older population of subjects. This should be commented on in the Discussion section of the manuscript.

Author Response

Dear Editors:

please see the attachment. Thanks for your revision

Shih-Lung Cheng

Reviewer 2 Report

I have read the article by Ching-Hsiung Lin et al. with great interest. The authors used Spirobank SmartTM to diagnose COPD and the results were compared to spirometry.

Comments:

  • Please explain the abbreviations at their first appearance (i.e. USPSTF)
  • “Portable is small and cheap device which constitute a valid alternative to spirometry”. Please rephrase, I guess a word is missing after portable.
  • Do you have any data on childhood asthma? Did this subgroup affected the results?
  • Why bronchodilator has not been administered before the portable spirometry? This could a potential reason for discrepancies (i.e. false positive rate). Please, clarify.
  • Please, comment on the accuracy and repeatability criteria for the confirmational lung function measurement.
  • What is the inter-day repeatability of the Spirobank portable device?
  • Statistical analysis. Please, provide power calculations.
  • Correlation analyses between the portable and standard devices are inaccurate to use. Instead, please, provide Bland-Altman plots to understand the magnitude and chance of discrepancies.
  • Once the Bland-Altman plots are created, please discuss them rather than the correlation analyses.
  • Inter-day repeatability of lung function test even in a case of a clinical trial population is considerable (https://pubmed.ncbi.nlm.nih.gov/32547001/). Please, discuss this as a limitation by citing the relevant article.
  • “CAT score greater than 10 is defined as symptomatic.”. Please, rephrase.

Author Response

Dear Editor:

please see the attachment. Thanks for your revision.

Shih-Lung Cheng

Round 2

Reviewer 2 Report

I was happy to see that there was a considerable improvement in this manuscript. However, there are still some aspects which have not been fully addressed.

  • I understand that patients were adults. But have you excluded patients who had asthma in childhood? They may have a higher variability in their lung function contributing to repeatability of the results.
  • I could not find the Bland-Altman plot in the manuscript. Please, ensure it is uploaded and inserted properly, so this vital figure can be assessed.

Author Response

Dear editor,

We would like to thank you for your kind letter and for reviewers’ constructive comments concerning our article. These comments are all valuable and helpful for improving our manuscript. All the authors have seriously discussed about all these comments. According to the reviewers’ comments, we have tried best to modify our manuscript to meet with the requirements of your journal. In this revised version, changes to our manuscript within the document were all highlighted by using red colored text. Point-by-point responses to the reviewers are listed below this letter.

Best Regards,

Shih-Lung Cheng
